# Dissecting fuel demand elasticities in Ghana: A quantile regression analysis using the Marshallian demand framework

Felix Takyi [1]*, Anthony Adu-Asare Idun [1], Patrick Kwashie Akorsu [1], Peace Yawo Ametepi[1], Peterson Owusu Junior[1,2]

1 Department of Finance, School of Business, University of Cape Coast, Cape Coast, Ghana, 2 School of Construction Economics & Management, University of the Witwatersrand, Johannesburg, South Africa

* felixtakyi145@gmail.com

**Editor:** sunny narayan, Tecnológico de Monterrey, MEXICO

## Abstract

Our study investigates fuel demand elasticities in Ghana by assessing consumer responses to income and price changes using the Marshallian demand framework. Applying Ensemble Empirical Mode Decomposition (EEMD), Quantile Regression Analysis (QRA), and Quantile-on-Quantile Regression (QQR), our study captures nonlinear, frequency-dependent relationships in petrol demand. Our study employed a monthly frequency data from 2000–2022. Our results reveal that petrol acts as a normal good in the medium term, but becomes inferior for high-income consumers in the long term. Petrol and diesel demonstrate both complementary and substitutive behaviours across different time horizons and income levels. The findings challenge traditional assumptions of price inelasticity and uniform demand, highlighting heterogeneity in consumer responses. We recommend differentiated fuel pricing policies and targeted subsidies to protect vulnerable groups. We also encourage investment in alternative energy sources and fuel-saving technologies. These insights are particularly valuable for policymakers, energy regulators, and development economists seeking to enhance fuel efficiency, economic stability, and energy diversification in Ghana.

## Introduction

Fuel consumption plays a vital role in Ghana's economic and industrial development. As a developing nation with intense dependency on transport, agriculture, and manufacturing, fuel price changes have far-reaching implications for businesses, households, and economic development [1]. Ghana has experienced repeated fuel price volatility over the last twenty years, largely induced by variations in global crude oil prices, exchange rate depreciation, and policy interventions by the government, including subsidy removal and taxation adjustments [2]. Volatility in the fuel market has raised concerns about its impact on living costs, the choice of machinery, inflation

**Data availability statement:** All relevant data for this study are publicly available from the figshare repository (https://doi.org/10.6084/m9.figshare.30109300).

**Funding:** The author(s) received no specific funding for this work.

**Competing interests:** The authors have declared that no competing interests exist.

rates, and overall economic efficiency. Consequently, it is crucial to understand and empirically assess how consumers respond to changes in fuel prices. Such insights are essential for informed policymaking aimed at enhancing economic stability and promoting efficient fuel consumption.

Petrol is the major source of energy for commercial and private transport in Ghana and has been a key determinant of mobility and economic engagement [3]. The asymmetric reaction of petrol demand to income and price changes is an empirical enigma that deserves attention. Classical economic theory argues that fuel demand is price inelastic in the short run due to the lack of substitutes and its essential nature [4]; yet consumer behaviour shows responsiveness depending on income levels, car ownership, and differences between urban and rural areas. Also, the relationship between petrol demand and diesel prices remains a debatable topic given the potential substitution between these two fuels in commercial transport sectors and industries. Petrol and diesel, both products of crude oil, nonetheless possess different chemical structures and ways of working.

Petrol creates a less dense fuel with a relatively lower boiling point, making it more appropriate for application in spark-ignition engines that are typical in personal cars and small transport fleets [5]. Diesel, on the other hand, is a denser fuel that needs compression ignition to become the most ideal in heavy transports, trucks, and industrial vehicles [5]. Diesel engines are generally perceived as being fuel-efficient and longer-lasting than petrol engines, and this results in varying patterns of demand for these two fuels. For Ghana, consumer choice may be characterised by the interaction of petrol and diesel prices, especially on the part of households, industries and owners of commercial transport who can alternate between the two fuels depending on relative price change and efficiency. Between 2020 and 2023, diesel prices in Ghana exhibited greater volatility, rising by nearly 200% compared to petrol's 150%, with both fuels priced below GHS5 per litre in 2020 and surpassing GHS13 per litre by 2024 [6]. Insights into such disparities are fundamental to examining the demand elasticities of petrol, as well as any prospective cross-price impacts with diesel.

The theoretical foundations of such an investigation rely on the Marshallian model of demand, according to which consumers' demand fluctuates based on relative price shifts and income fluctuations. The Marshallian approach provides a good foundation for analysis of demand elasticities insofar as isolating income and substitution effects is concerned [7]; however, it is inappropriate in capturing the nonlinear characteristics of economic relationships, which is a precondition for our analysis of demand dynamics. The theory also hypothesises uniform demand elasticities without accounting for heterogeneity of income and substitution effects across time horizons. The model also fails to account for heterogeneity in consumer responses, since it does not analyse the behaviour of demand at different points in the distribution, but relies on average impacts, ignoring the fact that different levels of consumers may react differently to different price levels. Further, the model's reliance on aggregate relations limits its ability to uncover differential patterns of demand in the short, medium, and long terms. Thus, while the Marshallian approach provides an essential insight into demand variability, it lacks the flexibility to capture multipart and evolving

demand setups with precision. As a result of these drawbacks in the model, this study seeks to utilise a non-linear and frequency-based model to capture the asymmetry responses of petrol demand as against different income and price changes by converting the linear Marshallian function into a non-linear model employing the quantile approach.

The study of fuel demand's associated elasticities with focus on changes in income and prices has employed a range of econometric approaches. Few studies have used the Marshallian demand model to evaluate the substitutability level between fuel and income elasticities [8–10], thus providing valuable intuitions into fuel usage patterns. These prior studies integrate major findings, methodologies used, and identified weaknesses within the current literature and highlights the value of the current study for fuel demand elasticities using quantile-based regression in the context of Ghana.

An early study on fuel demand elasticities in Pakistan by [8] employed the Linear Almost Ideal Demand System (LAIDS) model to estimate own-price as well as cross-price elasticities for electricity, gas, and other fuels. Their findings suggested that electricity and gas were substitutes, whereas electricity was found to be a complement to other alternative fuels. In addition, their study estimated expenditure elasticities, and it was found that all groups of fuel being studied were categorised as necessities, as their elasticities were less than unity. [10] analysed gasoline demand elasticities with a focus on the heterogeneous behaviour of different households in the United States. Their study used a translog demand function to capture variations in price and income elasticities across income groups. A striking observation in this study was the fuel price elasticities that showed a U-shaped pattern, which declined first with income and then increased at high income levels. In addition, [9] explored the energy consumption patterns of urban households in Ethiopia and identified price and income as the key determinants. The study revealed that the use of modern fuels rose with income, whereas the use of traditional fuels declined. The mentioned substitution effect confirms Engel's Law, which asserts that with a rise in income, households switch towards superior quality energy resources [11].

Recent studies have advanced the understanding of fuel demand elasticity by examining how consumers respond to price and income shocks across different countries and contexts. [12] investigated Iran's fuel subsidy reforms using a time-varying panel model with regional data, finding that fuel demand elasticity increases significantly after price reforms, especially in the long run, and that smuggling plays a key role in biasing traditional estimates. [13] analysed Mexican household survey data with QUAIDS and found demand for fuels to be generally price inelastic, with statistically significant differences between poor and non-poor households and income elasticities typically larger than price elasticities. Meanwhile, [14] used a Bayesian VAR framework on Italian data to show that fuel demand, particularly diesel, is rigid in the short run but more elastic over time, with strong sensitivity to Brent crude shocks. The models used in these studies; QUAIDS, time-varying panel models, and BVAR offer advantages such as flexibility in capturing dynamic responses, robustness to endogeneity [12], and realistic representation of fuel demand. However, their main limitation lies in estimating average effects that mask heterogeneity across the distribution of consumers and prices excluding [12], whose work targeted dynamic changes over time. Nonetheless, these methods assume homogeneous behaviour and often fail to reflect how different consumer segments respond to varying magnitudes of price shocks. The quantile approach addresses this gap by modelling the relationship between specific quantiles of consumption and price distributions, allowing for a grittier analysis of nonlinear and asymmetric responses. This makes quantile approaches particularly valuable for policy design, as it highlights where price changes have the strongest or weakest effects across the consumption spectrum.

The findings from prior literature highlight the need to use a frequency-quantile models in fuel demand analysis due to the heterogeneous nature of consumer behaviour even though [10] examined the heterogenous behaviours of consumer but failed to capture the frequency aspect. Since price and income elasticities varies, linear models may over-simplify these dynamics and thus fail to capture the varied responses adequately. High-income households are more responsive to price increases due to their discretionary fuel consumption [9], thus supporting the fact that elasticities are not uniform and require models that vary by income stratification as seen in the study of [10]. In addition, the studies unveil complex substitution and complementarity effects between fuels, which are better handled using non-linear approaches. From a

policy perspective, blanket measures can be ineffective if fuel demand responses differ by income group, and thus the need for targeted policies. Finally, using linear models to forecast fuel demand may lead to inaccuracies, especially during periods of economic uncertainty or price instability, while non-linear models can provide more accurate estimates. These factors recommend the use of non-linear modelling techniques to obtain more credible demand estimates and inform well-targeted policy formulation.

By applying this theoretical basis to the quantile-based regression models, our study attempts to shed light on the heterogeneity in responses exhibited by various segments of fuel consumers also in frequency-conditional bases. In contrast to the conventional mean regression approach based on a single response across the population, quantile regression (QR) and quantile-on-quantile regression (QQR) permits a more sophisticated analysis by investigating heterogeneity in demand elasticity across different points of the conditional distribution of petrol consumption and also considering the conditional distribution of the independent variables (diesel and petrol prices and income), respectively. To achieve the research objective, this study employs a QR and QQR together with the Ensemble Empirical Mode Decomposition (EEMD) method approach in estimating petrol demand elasticities in Ghana. This approach accounts for the varying sensitivities of low, middle, and high petrol consumers to price and income changes across different time periods. The study employs time series data on fuel prices (petrol and diesel) and consumer spending representing income based on the Marshallian demand framework, from the framework, the Cobb Douglas approach for calculating quantity demand will be utilise to ascertain the demand for petrol (see methodology section). This empirical approach provides deep insight into the frequency asymmetry of petrol demand elasticities and offers the required inputs to policymakers in formulating fuel price policy, tax structures, and subsidy programs according to the needs of consumers.

Ghana is selected as the sole focus of this study due to its pronounced fuel price volatility, which has been more significant than in many other countries. Between 2020 and 2022, petrol prices in Ghana surged by approximately 150%, while diesel prices increased by nearly 200%, largely driven by a 40.88% depreciation of the Ghanaian cedi against the US dollar in 2022. This volatility contrasts with other African nations, such as Egypt, which experienced minimal average changes of about 1.29% in fuel prices during the same period. Additionally, Ghana's fuel price fluctuations were more pronounced than those in oil-producing countries in Africa like Algeria and Libya with Ghana experiencing a 15.84% decrease in ex-pump prices between February and March 2022, compared to 1.79% in the Algeria and 7.81% in Libya. This significant and unique fluctuations in Ghana's fuel prices underscore the need for a focused analysis to understand consumer responses and inform targeted policy interventions.

Overall, results of this research will add to the wider debate on issues pertaining to energy economics, price policy effectiveness, and economic welfare in Ghana. This study will provide vital information on changes in fuel demand. It is the first study ever to examine income and substitution effects in Ghana, utilising petrol, diesel and consumer spending and such information is vital in knowing how fuel market consumers in Ghana behave. Second, this is the first application of EEMD, QR and QQR to examine how demand for petrol changes relative to its own price, price of diesel and income. The traditional approach is built around assuming that all consumers behave identically, which does not account for how various income classes react differently when income and prices change. Using EEMD, QR and QQR, this study illustrates the whole spectrum of consumer responses, providing a clearer idea of various fuel demands within the study period. Lastly, our study will suggest policy change beyond average effects to account for varied consumer responses to enable more targeted and equitable fuel pricing by identifying which groups are most sensitive to price changes, helping design reforms that are both effective and socially fair.

The results indicated distinct relationships between petrol demand, income, and fuel prices, reflecting key economic behaviours. A positive relationship was observed between petrol demand and income in the medium term, suggesting that petrol was a normal good, while a negative long-term relationship was evident for medium and high consumers, implying that petrol became an inferior good as income rose. Additionally, a negative relationship was found between petrol demand and diesel prices in the short and medium term, suggesting that petrol and diesel acted as complementary goods

during these periods, whereas the positive long-term relationship indicated that they functioned as substitutes over time. In addition, the negative short-term relationship between petrol demand and petrol prices suggested that petrol was price elastic in the short run, meaning consumers reduced consumption when prices rose. However, the positive relationship in the medium and long term indicated that petrol became price inelastic, reinforcing its necessity despite higher costs.

The subsequent sections provide a discussion of the research methodology, data and preliminary analysis, empirical analysis, and a conclusion which has policy and practical recommendations.

## Materials and methods

### Marshallian demand equation derivation

Understanding the demand for petrol involves a comprehensive investigation of consumer behaviour, particularly in response to changes in pricing and consumer spending. Consumers divide their financial resources between several fuel options, including petrol and its equivalent, diesel. Their purpose is to maximise satisfaction while maintaining within their budget restrictions. To formalise this decision-making process, we utilise a utility maximisation framework, which allows us to construct the Marshallian demand function for petrol.

The Marshallian demand function can be effectively derived using the Cobb-Douglas utility function, which assumes that consumer preferences follow a specific functional form. In this framework, a rational consumer seeks to maximise their utility, which is represented by the function:

$$U\left(X_1, X_2, \ldots, X_n = X_1^{\alpha_1} X_2^{\alpha_2} \ldots X_n^{\alpha_n}\right) \tag{1}$$

where each $\alpha_i$ represents the elasticity of utility with respect to fuel $X_i$. The sum of these elasticities is assumed to be equal to one, ensuring constant returns to scale. This assumption implies that the consumer distributes their expenditure in a way that maintains proportional consumption shares across different fuels.

To understand the consumer's decision-making process, we introduce the budget constraint, which limits the total expenditure on all fuel to the available income. Mathematically, this is expressed as

$$M = \sum_{i=1}^{n} P_i X_i \tag{2}$$

where $M$ denotes the total income, in our case consumer spending, $P_i$ represents the price of fuel $i$, and $X_i$ is the quantity consumed of fuel $i$. The consumer, therefore, faces a constrained optimisation problem where they must allocate their limited income to maximise utility.

To solve this problem, we employ the Lagrangian method, which incorporates the constraint into the objective function. The Lagrangian expression takes the form:

$$L = X_1^{\alpha_1} X_2^{\alpha_2} \ldots X_n^{\alpha_n} + \lambda \left(M - \sum_{i=1}^{n} P_i X_i\right) \tag{3}$$

where $\lambda$ is the Lagrange multiplier, representing the marginal utility of income. The first-order conditions for optimisation are obtained by differentiating the Lagrangian with respect to each fuel $X_i$ and the multiplier $\lambda$, then setting the derivatives equal to zero. This yields the condition:

$$\frac{\partial L}{\partial X_i} = \alpha_i X_i^{\alpha_i - 1} \prod_{j \neq i} X_j^{a_j} - \lambda P_i = 0, \ \forall_i \tag{4}$$

$$\frac{\partial L}{\partial \lambda} = M - \sum_{i=1}^{n} P_i X_i = 0 \tag{5}$$

Rearranging the first-order condition for $X_i$, we express the Lagrange multiplier in terms of the marginal utility and price:

$$\lambda = \frac{\alpha_i}{P_i} X_i^{\alpha_i - 1} \prod_{j \neq i} X_j^{\alpha_j} \tag{6}$$

To derive the Marshallian demand functions explicitly, we solve for by substituting back into the budget constraint. This results in:

$$X_i = \frac{\alpha_i M}{P_i} \tag{7}$$

which expresses the optimal quantity demanded of a fuel as a function of its price, the consumer's total income, and the parameters of the utility function. This demand function captures the fundamental economic principle that consumption depends on both income and price levels.

A critical insight from this derivation is the decomposition of the consumer's response to price and income changes into the income effect and the substitution effect. The income effect describes how an increase in income influences consumption. In the case of a normal good, demand increases as income rises $\left(\frac{\partial X_i}{\partial M} > 0\right)$, whereas for an inferior good, demand falls as income increases $\left(\frac{\partial X_i}{\partial M} < 0\right)$. The substitution effect, on the other hand, captures how changes in the price of one good influence the demand for another. If two goods are substitutes, an increase in the price of one lead to higher demand for the other $\left(\frac{\partial X_i}{\partial P_j} > 0\right)$. Conversely, if the goods are complements, an increase in the price of one reduces the demand for the other $\left(\frac{\partial X_i}{\partial P_j} < 0\right)$. Additionally, the own-price effect states that, in accordance with the law of demand, an increase in the price of a good generally leads to a decrease in its demand $\left(\frac{\partial X_i}{\partial P_i} < 0\right)$.

Empirical estimation of the Marshallian demand function often involves transforming the Cobb-Douglas equation into a log-linear form for econometric analysis. This transformation takes the form:

$$lnX_i = ln\alpha_i + lnM - lnP_i + \epsilon \tag{8}$$

where $\epsilon$ represents a stochastic error term. Traditional econometric techniques, such as Ordinary Least Squares (OLS) regression, estimate the mean effect of price and income on demand. However, more sophisticated methods, such as quantile-based regression, provide deeper insights into consumer behaviour across different price and income groups, highlighting heterogeneity in price and income elasticities.

## Model Estimation

This study partly reproduces the methods employed by [15,16], and [17]. The main idea is to investigate the asymmetric relationship between the demand for Petrol and prices of diesel, petrol and income. The study incorporates data that reflects the diverse preferences of investors. In achieving this, a two-step methodological approach is employed, integrating Ensemble Empirical Mode Decomposition (EEMD), Quantile Regression Analysis (QRA), and Quantile-in-Quantile Regression (QQR). The EEMD technique is utilised to mitigate noise and extract Intrinsic Mode Functions (IMFs) from our variables (demand for Petrol, Price for Petrol and Diesel and consumer spending), each corresponding to distinct time horizons. These IMFs capture different time scales within the original time series, while QRA and QQR address asymmetries response in demand for Petrol. This combined approach is instrumental in analysing the complex stylised facts of the time series, which are crucial to this investigation [18]. EEMD operates by decomposing the original series through multiple iterations, during which white noise is introduced to provide a consistent reference in the frequency domain [19].

## Ensemble empirical mode decomposition (EEMD)

As an advancement over Empirical Mode Decomposition (EMD), EEMD enhances the original EMD objective by further minimising noise. The methodology outlined by [19] is adopted and implemented in this study.

The EEMD defines the IMF components as the mean of an ensemble of trials, where each is made of the signal (data) and white noise of finite amplitude. In generic terms, all data $x(t)$ are a sum of signal (i.e., actual data, $s(t)$) and noise $n(t)$ so that

$$x(t) = s(t) + n(t) \tag{9}$$

EEMD introduces white noise to suppress weak signals while preserving the true underlying signal. This approach is inspired by the works of [20] and [21]. The principle of cancellation effects, applied across multiple noise-added cases, enhances the accuracy of results. According to Eq. (9), an $i^{th}$ synthetic observation $x_i(t)$ in Eq. (10) is generated by incorporating a white noise realisation $w_i(t)$. This process prevents mode mixing by maintaining a relatively uniform reference scale distribution, thereby facilitating effective EMD.

$$x_i(t) = x(t) + \omega_i(t) \tag{10}$$

The development of EEMD relies on the properties of EMD of [22] and [23] in the following manner: add white noise to the targeted data to arrive at $x_i(t)$, decompose $x_i(t)$ into IMFs, iterate 1 and 2 with varying white noise series and obtain the (ensemble) means of corresponding IMFs of the decomposition as the result.

The important final feature is that the mean IMFs reside within the natural dyadic filter windows to evades the mode mixing problem. The largest number of IMFs $s_i$ (and one residual $r$) of a data set is approximately $log_2 N$ where $N$ represents the total number of data points. Thus, $r$ can be represented as $s_i - (s_i - 1)$. The most compelling advantage of the EEMD specifically addresses the mode mixing problem that plagued the original EMD. By adding white noise realisations and ensemble averaging, EEMD provides more meaningful IMFs that better separate different oscillatory components. Other methods like the wavelet decomposition, while avoiding mode mixing per se, can suffer from spectral leakage between adjacent frequency bands, particularly when analysing signals with closely spaced frequency components or rapid frequency variations. Also, EEMD maintains superior temporal localisation throughout the decomposition process, as each IMF preserves the original temporal resolution of the input signal, whiles other decomposition methods down sample at each decomposition level, potentially losing fine temporal details at higher frequency components.

## Quantile-in-Quantile Regression (QQR)

The QQR technique serves as the non-parametric counterpart to QRA. Since it does not rely on parameters, there is no direct way to assess the adequacy of its estimates. Therefore, an empirical comparison between QRA and QQR provides justification for the conditional quantile relationship between the variables. This makes the QQR approach particularly suitable for analysing the increasing or decreasing relationship between petrol demand and its price, as well as the price of diesel and income (consumer spending). By capturing quantiles, this method effectively highlights the asymmetry between high and low patterns of price, income, and demand. The analysis begins with the following nexus:

$$DP_t^\theta = \beta (PI_t) + u_t^\theta \tag{11}$$

where $DP_t$ denotes the percentage differenced of demand for petrol and $PI_t$ denote the percentage differenced price of petrol/diesel/income at period $t$, $\theta$ is the $\theta th$ quantile of the conditional distribution of $DP_t$ and $u_t^\theta$ is the error quantile

whose $\theta th$ conditional quantile is made up to be zero, and $\beta(.)$ represents the slope of this relationship. By expanding [11] through the order Taylor rule, the quantile of $EN^\tau$ is given as:

$$\beta^\tau (PI_t) \approx \beta^\theta (PI^\tau) + \beta^{\theta'}(PI^\tau)(PI_t - PI^\tau) \tag{12}$$

Where $\beta^{\theta'}$ explains the partial derivative of $\beta^\theta(PI_t)$ which represent the marginal effect. We see that $\theta$ is the functional form of $\beta^\theta(PI^\tau)$ and $\beta^{\theta'}(PI^\tau)$ while $\tau$ is the functional form of $PI$ and $PI^\tau$, hence $\theta$ and $\tau$ are the functional forms of $\beta^\theta(PI^\tau)$ and $\beta^{\theta'}(PI^\tau)$. If we represent $\beta^\theta(PI^\tau)$ and $\beta^{\theta'}(PI^\tau)$ by $\beta_0(\theta, \tau)$ and $\beta_1(\theta, \tau)$, respectively, then [13] can suffice:

$$\beta^\theta (PI_t) \approx \beta_0(\theta, \tau) + \beta_1(\theta, \tau)(PI_t - PI^\tau). \tag{13}$$

By substituting [13] into [11], we arrive at the [14] as:

$$DP_t = \frac{\beta_0(\theta,, \tau) + \beta_1(\theta, \tau)(PI_t - PI^\tau)}{(*)} + u_t^\theta \tag{14}$$

Where ($^*$) gives the conditional quantile of $\theta th$ of percentage difference in demand for petrol. Furthermore, it reveals the actual association between the quantile of prices of petrol/diesel/income ($\tau th$) and the quantile of percentage difference of demand for petrol ($\theta th$) of parameters $\beta_0$ and $\beta_1$ with indices of $\theta$ and $\tau$. A similar minimisation in the ordinary least squares framework results in:

$$\min_{b_0 b_1} \sum_{i=1}^{n} \rho\theta \left[ PI_t - b_0 b_1 \left( \widehat{PI_t} - \widehat{PI^\tau} \right) \right] K \left( \frac{F_n \left( \widehat{PI_t} \right) - \tau}{h} \right) \tag{15}$$

where $\rho_\theta(u)$ is the quantile loss function represented as $\rho\theta(u)=u(\theta - I(u<0)) < \rho_\theta(u) = u(\theta - I(u < 0))$, $i$ is the function of indicator, $K(.)$ is the kernel density function and $h$ denotes kernel density function bandwidth parameter. The kernel function weights the observations of $PI\tau\ PI^\tau$ where the minimal weights are negatively related to the distribution function of $\widehat{PI_t}$ as $F_n \left( \widehat{PI_t} \right) = \frac{1}{n}\sum_{k=1}^{n} \left( \widehat{PI_k} < \widehat{PI_t} \right)$. Given the limited number of observations due to monthly data usage, it is essential to select an appropriate number of quantiles that ensures a minimum of 30 observations per estimation in each quantile to meet the statistical requirements for sample size, as a result of that, the study utilised 7 quantiles since we have 275 data points.

By using QRA and QQR and IMFs, we are able to quantify the relationship between demand for petrol, income, prices for petrol and diesel in a way that captures their frequency-dependent and non-linear link. Furthermore, this allows us to infer the nexus during upward and downward episodes in the short-, medium-and long-terms.

## Data

The study utilised monthly frequency data for all variables. The selection of these variables is based on their variability in prices and usage of these fuels in the country. Additionally, the sample period for the study was from January 2000 to December 2022, as they were determined by data availability rather than the full historical records in the country. Data were sourced from Bank of Ghana and Ghana statistical service website. The original series were converted into returns $r = \left( \frac{p_t}{p_0} - 1 \right)$ to align with stylised facts to meet the stationary feature and facilitate the use of subsequent econometric models to be used.

The study used the Marshallian demand framework, however used EEMD decomposition technique to capture the frequency-based response dynamics in the fuel market in Ghana. A key strength of EEMD is its ability to endogenously

decompose a series into multiple IMFs based on the series' length, capturing different time horizons. In the literature, IMF 1, IMF 5, and the residual IMF are commonly used to represent short-, medium-, and long-term dynamics [17], respectively. As such, IMF 1 represents short-term dynamics, capturing immediate consumer responses to price changes, seasonal variations, and monthly fluctuations where behavioural adjustments are limited by existing constraints. IMF 5 reflects medium-term structural adjustments spanning several time periods, encompassing business cycles, policy implementation effects, and moderate consumer adaptations such as vehicle choice changes or transportation mode shifts. The residual IMF also captures long-term trends and fundamental structural changes, including urbanisation effects, economic development trajectories, and generational behavioural shifts that fundamentally alter fuel consumption patterns.

For the purpose of this study, we utilised seven quantiles (Q); Q0.05 and Q0.20 represent low petrol consumers, low fuel prices, and low-income earners; Q0.35 to Q0.65 correspond to middle petrol consumers, normal fuel prices, and average-income earners; and Q0.80 to Q0.95 represent high petrol consumers, high fuel prices, and high-income earners. The selection of seven quantiles is grounded in statistical power considerations given the study's sample size of 275 monthly observations. QR requires adequate sample density within each conditional quantile to ensure reliable parameter estimation, and with 275 data points, seven quantiles provide approximately 39 observations per quantile segment, which meets the minimum threshold for stable coefficient estimation while avoiding small-sample bias that would emerge with more quantiles. Excessive quantile partitioning would lead to unstable estimates due to insufficient observations in the tail quantiles, while fewer than seven quantiles would compromise the study's ability to capture heterogeneous fuel demand responses across the conditional distribution, potentially masking critical variations in elasticity estimates. Therefore, the seven-quantile specification represents the optimal methodological approach that maximizes informational content while maintaining the statistical rigor necessary for reliable inference in QR analysis. The next section presents the summaries of our variables.

## Results and discussion

### Descriptive Statistics

Table 1 presents the summary statistics of the return series of our variables. The mean quantity demanded for petrol is the highest, indicating that petrol is a frequently consumed commodity. Additionally, petrol exhibited the highest standard deviation, suggesting that its demand dynamics are influenced by various factors, in our case price and income fluctuations. The normality test results indicate that the variables are not normally distributed, while the stationarity test confirms that most variables are stationary. However, the non-stationarity observed in the residuals for demand for petrol and price of diesel raises concerns about the reliability of estimated elasticity relationships, they suggest that the estimated fuel demand-price relationships may not be stable over the 275-month study period, potentially indicating structural breaks or time-varying elasticity parameters. This to some extent can significantly affect policy conclusions by making long-term forecasts unreliable and potentially rendering policy recommendations regarding fuel taxation or subsidy reforms ineffective. These findings justify the use of a non-linear model to effectively capture the asymmetric behaviour of petrol, diesel, and income.

### Empirical analysis

### QR regression results

The QRA estimates the conditional quantiles of the quantity demanded for petrol on the conditional averages of petrol and diesel prices and income levels. The aim is to capture the dynamic response of petrol demand to changes in average income and price levels. This approach helps to assess income and substitution effects across different quantiles and time

**Table 1. Summary of the series.**

| Stat | Demand for Petrol | | | | Consumer Spending | | | |
|---|---|---|---|---|---|---|---|---|
| | Original | IMF 1 | IMF 5 | Residual | Original | IMF 1 | IMF 5 | Residual |
| Obs. | 275 | 275 | 275 | 275 | 275 | 275 | 275 | 275 |
| Min | −0.9631 | −16.8679 | −1.2608 | −0.5855 | −0.7049 | −2.8460 | −0.1107 | −0.0236 |
| Max | 35.4193 | 15.2307 | 2.3308 | 4.7675 | 6.6902 | 2.8946 | 0.1555 | 0.0168 |
| Mean | 0.9719 | −0.1344 | −0.0210 | 0.9057 | 0.0191 | 0.0203 | 0.0054 | −0.0134 |
| Std. Dev. | 4.3002 | 3.6695 | 0.6710 | 1.5971 | 0.4135 | 0.3647 | 0.0498 | 0.0109 |
| Normtest W | 0.3428*** | 0.7762*** | 0.9641*** | 0.8417*** | 0.1028*** | 0.3817*** | 0.8426*** | 0.8340*** |
| ADF | −4.9203*** | −8.1601*** | −3.0403** | 0.0545 | −6.5517*** | −11.001*** | −5.6887*** | −11.371*** |
| | Price of Diesel | | | | Price of Petrol | | | |
| | Original | IMF 1 | IMF 5 | Residual | Original | IMF 1 | IMF 5 | Residual |
| Obs. | 275 | 275 | 275 | 275 | 275 | 275 | 275 | 275 |
| Min | −0.9902 | −9.5425 | −0.8477 | −0.2710 | −0.9722 | −12.2500 | −0.9890 | −0.2883 |
| Max | 23.0000 | 9.0732 | 0.8434 | 1.6413 | 26.0000 | 10.9648 | 1.3517 | 1.8823 |
| Mean | 0.6657 | −0.0045 | −0.0074 | 0.7615 | 0.7994 | −0.0881 | 0.0207 | 0.7824 |
| Std. Dev. | 2.7870 | 2.2368 | 0.3857 | 0.5562 | 3.1298 | 2.5249 | 0.4999 | 0.6301 |
| Normtest W | 0.4032*** | 0.8356*** | 0.9878*** | 0.9509*** | 0.4257*** | 0.8491*** | 0.9782*** | 0.9545*** |
| ADF | −5.4752*** | −7.757*** | −3.8839*** | 2.2918 | −5.0453*** | −7.41*** | −4.1289*** | −1.3346 |

Note: Normtest.W* indicates Shapiro-Wilk test of normality and ADF- Augmented Dicker-Fuller which are rejected for all levels of significance.

**Table 2. Quantile regression analysis.**

| Quantiles | Consumer Spending | | | Price of Diesel | | | Price of Petrol | | |
|---|---|---|---|---|---|---|---|---|---|
| | IMF 1 | IMF 5 | Residual | IMF 1 | IMF 5 | Residual | IMF 1 | IMF 5 | Residual |
| 0.05 | 0.2692 | 3.1241*** | 18.3491*** | −0.0001 | −0.6601*** | −0.7681*** | −0.4726*** | 0.3222 | −0.5217*** |
| 0.20 | 0.3673 | 3.1397*** | −5.07435 | −0.0001 | −0.6601*** | 0.3918*** | −0.4726*** | 0.3418* | 0.5784*** |
| 0.35 | 0.4189 | 3.1547*** | −24.0015*** | −0.0001 | −0.6608*** | 1.0288*** | −0.4726*** | 0.3989** | 1.1446*** |
| 0.50 | 0.5441** | 3.1682*** | −41.3532*** | −0.0001 | −0.6608*** | 1.5292*** | −0.4726*** | 0.4001*** | 1.5459*** |
| 0.65 | 0.8141*** | 3.2082*** | −65.5128*** | −0.0001 | −0.6668*** | 1.9734*** | −0.4726*** | 0.4510*** | 1.8811*** |
| 0.80 | 0.8141*** | 3.2354*** | −104.463*** | −0.0001 | −0.6700*** | 2.3671*** | −0.4726*** | 0.4545*** | 2.1753*** |
| 0.95 | 0.8141*** | 3.2982*** | −181.935*** | −0.0001 | −0.6700*** | 2.7493*** | −0.4726*** | 0.4846*** | 2.4337*** |

Note: Entries with (***) p < 0.01, (**) p < 0.05, and (*) p < 0.1 respectively. 0.05 and 0.20, 0.35–0.65, and 0.80 and 0.95 represents low, medium and high petrol consumers, respectively.

horizons, considering various IMFs. This addresses the limitations of the Marshallian demand framework, which assumes a linear response of demand to price and income changes (Table 2).

Our results show that in the short-term lower quantiles, the relationship between petrol demand and consumer spending is positive but insignificant. However, a significant positive relationship is observed in the middle to upper quantiles (50th to 95th quantiles). This suggests that low petrol consumers, such as households, are unresponsive to income changes, as indicated by the insignificant coefficients. In contrast, for middle and high petrol consumers, petrol functions as a normal good. In the medium term, the positive relationship persists, but this time, all quantiles are significant, implying that income increases lead to higher petrol demand across all consumer categories, emphasising the necessity of petrol for their activities, this results is consistent with the findings of [8,9] where their study found fuel to be a normal good but considered only a single time period. Conversely, in the long term, a negative association is found for medium

and high petrol consumers, indicating a reduction in petrol demand as income rises. This suggests that these consumers opt for better fuel alternatives, rendering petrol an inferior good in the long run. However, low petrol consumers maintain a positive relationship with income, suggesting that their demand for petrol continues to rise despite long-term income growth.

For the price of diesel, we observed a negative relationship with the quantity demanded for petrol in both the short and medium term, though the coefficient was insignificant in the short term. The significant negative relationship in the medium term suggests that when the price of diesel increases, the quantity demanded for petrol decreases. This finding is counter-intuitive because petrol and diesel are generally considered substitutes, but not too surprising as [8] also found gas to be a complementary good to other fuels. Typically, when diesel prices rise, consumers should switch to petrol. However, our results suggest otherwise, revealing that petrol and diesel act as complementary goods in this context. In the long term, the relationship reversed, showing a positive relationship from the 20th to the 95th quantile, while the lowest quantile (5th quantile) still exhibited a significant negative relationship. The positive relationship suggests that when the price of diesel rises, the demand for petrol also increases, reflecting their substitutive nature as fuel commodities. However, for low-consumption groups, petrol and diesel remained complementary.

Lastly, the study examined the influence of petrol prices, in line with Marshallian demand theory, where demand is a function of its own price. A significant association was observed across all quantiles and time horizons, but with varying directions. The short term showed a negative relationship, while the medium- and long-term exhibited a positive relationship. In the short term, the negative relationship indicates that when petrol prices increase, consumers across all categories reduce their petrol consumption and switch to alternatives, demonstrating its elastic characteristics during this time. However, this relationship reversed in the medium and long term, where demand for petrol increased for all consumers despite rising prices, reflecting the inelastic nature of petrol consumption over time, supporting the findings of [13] where they found petrol to be an inelastic commodity. Notably, low petrol consumers continued to buy less petrol as prices increased, as indicated by the negative coefficient in the lower quantiles.

This result helps address the limitations of Marshallian demand theory, as our study considers the asymmetric responses of different types of petrol consumers as a function of average change in prices and income. However, [24] identified limitations in the use of QR, which necessitated the use of QQR. Unlike QR, QQR decomposes the independent variables into quantiles and regresses them against the quantiles of the dependent variable. In our case, the method breaks down petrol and diesel prices, as well as income into quantiles and regresses them against the quantiles of petrol demand. Table 3 presents the results of the QQR estimates.

## QQR estimates

Table 3 presents the cross-sectional averages of the QQR, as postulated by [24]. It is important to note that QQR is a nonparametric model and, therefore, does not provide significance values. However, to validate its significance, we

**Table 3. QQR estimates.**

| Quantiles | Consumer Spending | | | Price of Diesel | | | Price of Petrol | | |
|---|---|---|---|---|---|---|---|---|---|
| | IMF 1 | IMF 5 | Residual | IMF 1 | IMF 5 | Residual | IMF 1 | IMF 5 | Residual |
| 0.05 | 0 | 0.0026 | 0.0041 | 0 | 0.0916 | 0.1505 | 0 | 3.8031 | 0.2115 |
| 0.20 | 1.0296 | −0.0079 | −0.0049 | −4.2321 | −0.2527 | 0.3651 | 27.8224 | −0.937 | 0.4161 |
| 0.35 | −0.0901 | 0.0333 | 0.0003 | 4.9450 | −0.4614 | 0.2377 | −14.9335 | 0.4583 | 0.2944 |
| 0.50 | −0.1456 | 0.0917 | 0.0026 | −2.9864 | −0.3032 | 0.1829 | −4.7488 | −0.3479 | 0.2420 |
| 0.65 | 0.2849 | −0.0006 | 0.0039 | 4.3974 | −0.2695 | 0.1504 | −16.4427 | 0.2219 | 0.2109 |
| 0.80 | 0.0655 | 0.0353 | 0.0049 | 25.3086 | 0.9268 | 0.1281 | 29.3448 | 2.9742 | 0.1895 |
| 0.95 | −0.1206 | 0.0066 | 0.0054 | −4.3762 | −0.3789 | 0.1138 | 5.6842 | 0.7511 | 0.1758 |

examine how closely its coefficients align with those of QRA. This validation is observed through the line plots of QRA and QQR coefficients in Fig 1.

The relationship is evident in the line plots of QRA and QQR coefficients presented in Fig 1. The results indicate that QRA estimates largely align with those of QQR across different quantiles and time horizons. Additionally, the magnitude and direction of QQR estimates closely match those of QRA, with a few exceptions observed in the long term. This deviation in the coefficients can be attributed to non-stationarity found in the residuals, which led to significant differences in coefficient estimation. Nonetheless, QQR can be said to be an effective method for capturing asymmetric relationships between petrol demand, petrol and diesel prices, and income, with estimates that are mostly well-aligned with those from QRA.

Finally, we present QQR estimates in three dimensions, as illustrated in Fig. 2. These plots display the QQR estimates of the slope coefficients as a function of the quantile parameters for both the dependent and explanatory variables. Specifically, the plots depict the slope coefficient on the z-axis, plotted against the quantile of petrol and diesel prices and income on the x-axis and the quantile of the quantity demanded of petrol on the y-axis. The results reveal nonlinear coefficient variations across quantiles.

### QQR 3-dimensional plots

Fig 2 displays the 3-dimensional plots of the QQR coefficients, which generally show results consistent with the QR estimates, with a few exceptions. For consumer spending (income), the colour gradients indicate a very weak negative association with petrol demand across all quantiles and frequencies. However, a noticeably strong negative relationship is observed among middle petrol consumers when income declines suggesting inferior good. For the price of diesel, the relationship varies dynamically across quantiles, but the most notable findings are as follows; the strong negative (positive) relationship between low (high) petrol consumers across all levels of price for diesel, making petrol a substitute for high consumers and compliment good for low consumers in the short run.

In the long run, this relationship persists, although the magnitude of association for high petrol consumers decreases. The relationship between the price and demand for petrol follows a similar pattern to that of diesel prices, confirming that price changes affect quantity demanded in comparable ways. The plot further validates the short-term elasticity of petrol for all categories of consumers in Ghana, whereas in the medium term, varying elasticities are observed. In the long term, petrol remains elastic for low consumers but inelastic for middle and high consumers. Overall, the results mostly align with the QRA estimates but offer a more detailed view of the true relationships across all quantiles of both dependent and independent variables.

### Robustness

Following the studies by [25] and [26], we employed the nonlinear causality test (causality in quantiles) proposed by Jeong et al. in 2012 and later improved by [27]. The results, as shown in Fig 3, are consistent with the QR estimates. Specifically, the findings highlight the causal role played by diesel and petrol prices, as well as income, in determining petrol demand. Notably, we can assess the significance of the quantile coefficients by comparing them against the causality test, where coefficients are considered significant when they exceed a t-statistic of 2 in the causality plots. As a result, we document overwhelming similarities and conclude that our findings are robust to an alternative method that accounts for nonlinearities in the relationships.

### Conclusion

This paper provides an in-depth assessment of petrol demand elasticities in Ghana, covering the asymmetric responses of consumers to changes in petrol and diesel prices, as well as income levels in frequency bases. By adding EEMD, QRA

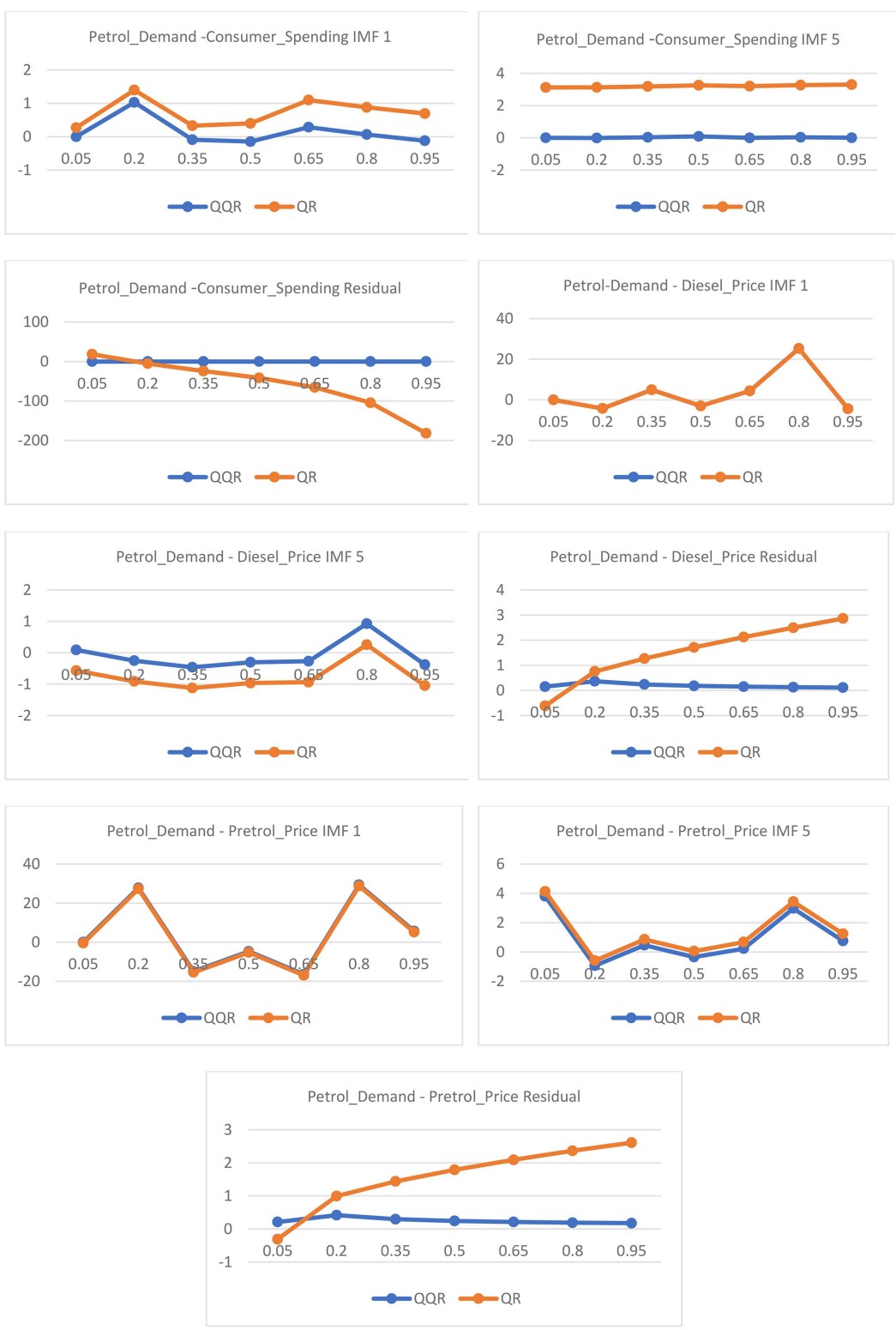

**Fig 1. QR and QQR coefficient plots of the quantity demanded, price of petrol and diesel and income.**

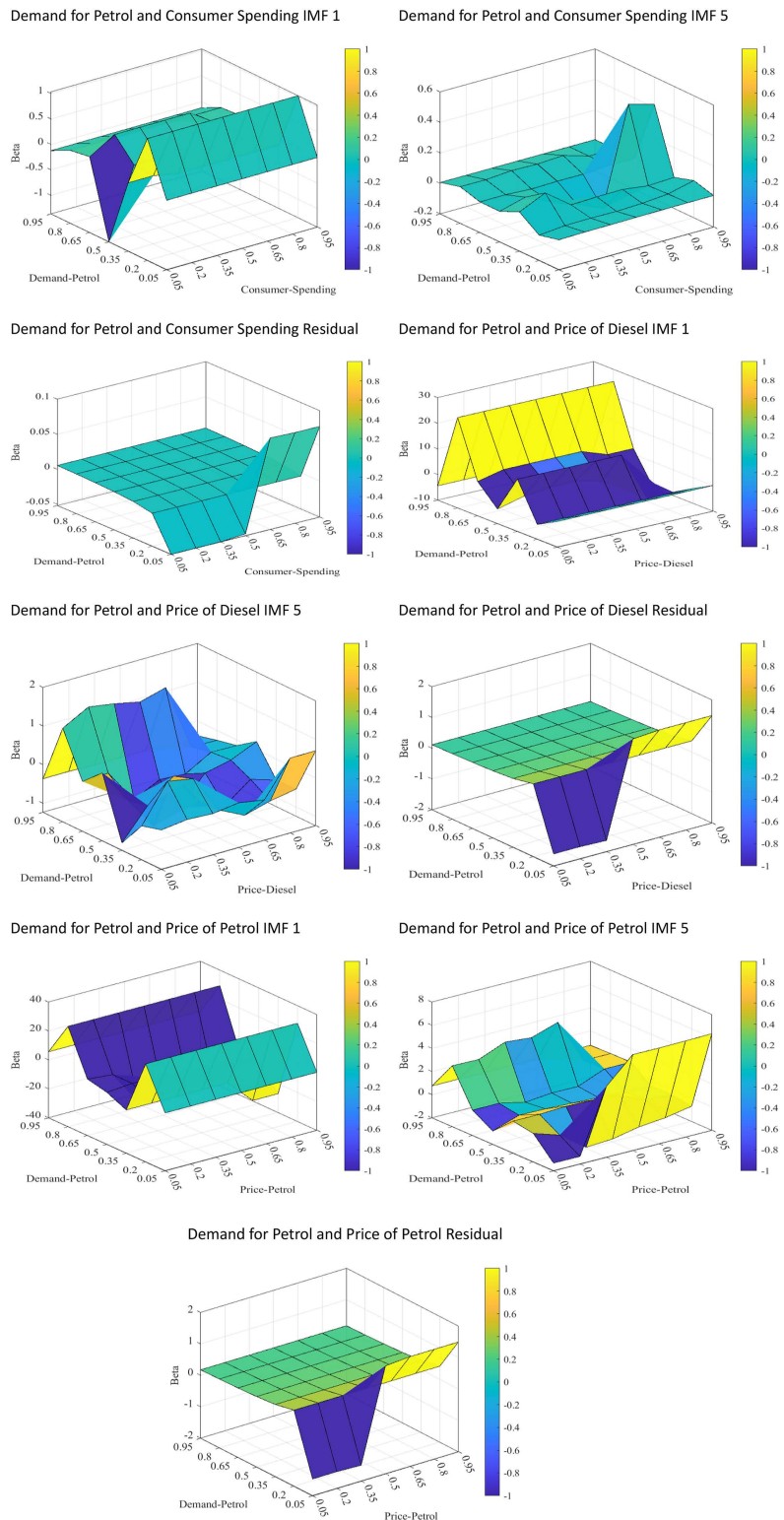

**Fig 2. QQR coefficient of quantity demanded for petrol, petrol and diesel prices and income.**

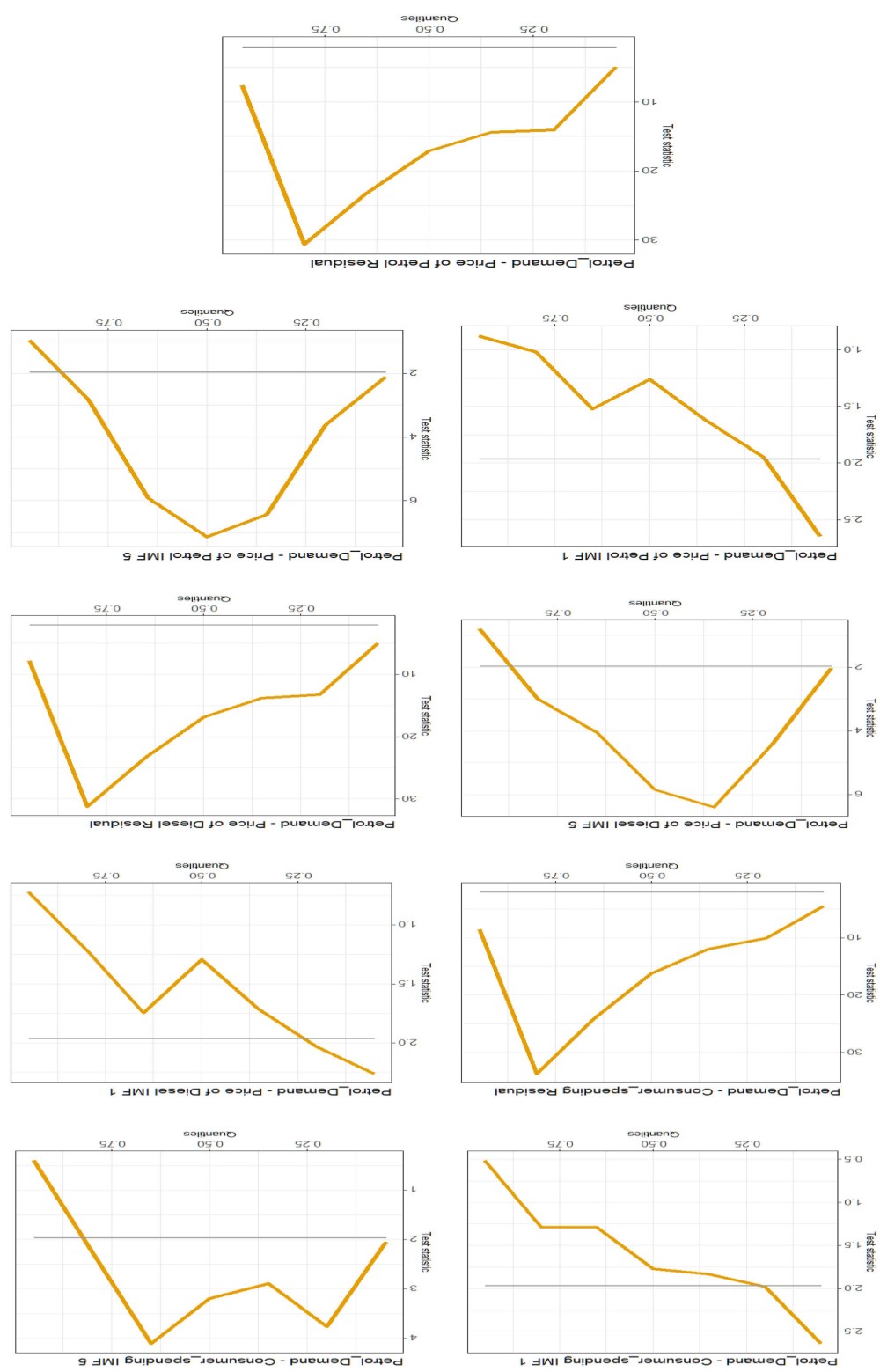

**Fig 3. Plots of causality-in-mean tests results.**

and QQR, the study advances beyond classic Marshallian demand models to capture the heterogeneity in consumer behaviour across different income groups and time horizons, given the increasing variability in fuel prices in Ghana. The findings underscore the complexity of fuel demand patterns, delivering relevant insights for policymakers, energy economists, and industry regulators. A fundamental result of this study is the need of fuel pricing systems that account for variations in customer responsiveness to price and income changes. The results underline that petrol demand is not uniform across consumers; instead, demand responses fluctuate dependent on income levels, consumption patterns, and the time horizon under consideration. This shows that blanket fuel pricing plans may not be effective

Furthermore, the study's findings undermine the premise that petrol and diesel are perfect alternatives. The observed complimentary relationship between petrol and diesel in some quantiles implies that increases in diesel pricing may not necessarily lead to a straight substitution effect with fuel. This calls for a reassessment of energy policy frameworks that presume a simple price substitution mechanism between these two fuel kinds. In light of this, energy planners should work on broadening the transport energy mix, promoting fuel economy measures, and investing in alternative energy sources such as electric vehicles or biofuels to minimise dependency on petrol and diesel.

To enhance fuel pricing policy, energy market stability, and demand forecasting in Ghana, policymakers should implement differentiated fuel taxation and subsidies to protect lower-income groups from disproportionate price increases. The government could implement income-tiered fuel pricing through digital subsidy systems linked to national ID cards basically the Ghana card, where low-income households receive allocated subsidised fuel while higher earners pay market rates, or through progressive taxation where initial monthly allocations are subsidised with subsequent purchases taxed at increasing rates.Also, Investment in alternative energy sources, such as electric transport and biofuels, should be prioritised to reduce reliance on petrol and diesel while promoting energy diversification. This can be addressed through public-private charging network partnerships, import duty exemptions, and grid improvements. Public awareness campaigns should be intensified to encourage fuel-saving behaviours, such as carpooling and public transport use. Additionally, expanding multi-fuel vehicle technologies and infrastructure for alternative fuels will provide consumers with greater flexibility. Ghana can also emulate Kenya's successful implementation of a 10% ethanol blending mandate combined with comprehensive clean energy tax incentives by adopting a phased approach that integrates subsidy reforms with robust social safety nets. This strategy should incorporate regional biofuel cooperation frameworks that leverage the country's agricultural resources and shared technological expertise while addressing Ghana's specific infrastructural limitations and economic constraints. These measures will enhance energy security, economic stability, and environmental sustainability, ensuring that fuel pricing aligns with consumer behaviour and economic realities in Ghana. By applying nonlinear approaches, our work provides a more precise and granular understanding of fuel demand patterns with the Marshallian demand theory.

## Limitations and suggestion for future research

This study is limited by its exclusion of broader macroeconomic variables such as exchange rates and global oil prices, which may influence fuel demand. Exchange rate fluctuations present a particularly critical limitation, as Ghana's reliance on imported petroleum products even though as an exporting country, means currency depreciation affects domestic fuel prices independently of global oil movements, potentially confounding the estimated price elasticity coefficients by capturing currency-induced price variations rather than pure demand responsiveness. Global oil price dynamics may also bias elasticity estimates if international price volatility exhibits different temporal patterns than domestic market adjustments. The study also does not account for demographic or behavioural factors like vehicle ownership or urbanisation, which may mask heterogeneous demand responses across population segments, as urban consumers typically exhibit different elasticity patterns due to alternative transportation availability, while vehicle ownership directly influences fuel demand necessity. These exclusions may result in elasticity estimates that reflect aggregate behavioural responses rather than underlying structural demand relationships, potentially creating omitted variable bias and limiting policy relevance. Future

research should integrate these variables and consider regional or cross-country comparisons to enhance generalisability and provide more robust estimates that account for the complex factors influencing energy consumption behaviours.

## Author contributions

**Conceptualization:** Felix Takyi, Peterson Owusu Junior.

**Formal analysis:** Felix Takyi.

**Methodology:** Felix Takyi, Anthony Adu-Asare Idun, Peterson Owusu Junior.

**Supervision:** Anthony Adu-Asare Idun, Patrick Kwashie Akorsu, Peace Yawo Ametepi.

**Validation:** Anthony Adu-Asare Idun.

**Visualization:** Felix Takyi.

**Writing – original draft:** Felix Takyi.

**Writing – review & editing:** Anthony Adu-Asare Idun.

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
