## [Decision Letter · Decision Letter 0]

3 Aug 2025

Dear Dr. Takyi,

Thank you for submitting your manuscript to PLOS ONE. After careful consideration, we feel that it has merit but does not fully meet PLOS ONE’s publication criteria as it currently stands. Therefore, we invite you to submit a revised version of the manuscript that addresses the points raised during the review process.

We look forward to receiving your revised manuscript.

Kind regards,

Sunny Narayan

Academic Editor

PLOS ONE

Journal Requirements:

2. In the online submission form, you indicated that “The data underlying the results presented in the study are available upon request”

freely available to other researchers, either 1. In a public repository, 2. Within the manuscript itself, or 3. Uploaded as supplementary information.

4. We note you have included a table to which you do not refer in the text of your manuscript. Please ensure that you refer to Table 2 in your text; if accepted, production will need this reference to link the reader to the Table.

Reviewers' comments:

Reviewer's Responses to Questions

**Comments to the Author**

1. Is the manuscript technically sound, and do the data support the conclusions?

Reviewer #1: Yes

2. Has the statistical analysis been performed appropriately and rigorously?

Reviewer #1: Yes

3. Have the authors made all data underlying the findings in their manuscript fully available?

Reviewer #1: Yes

4. Is the manuscript presented in an intelligible fashion and written in standard English?

Reviewer #1: Yes

Reviewer #1: Review of Manuscript

Title: Dissecting Fuel Demand Elasticities in Ghana: A Quantile Regression Analysis Using the Marshallian Demand Framework

The manuscript presents a rigorous analysis of fuel demand elasticities in Ghana, employing advanced econometric techniques to explore nonlinear and heterogeneous consumer responses to price and income changes. The study is well-structured and methodologically sound, offering valuable insights for policymakers and energy economists. However, certain areas could benefit from clarification, expansion, or refinement to enhance the manuscript's impact and readability.

Strengths:

1. Methodological Rigor:

o The use of Ensemble Empirical Mode Decomposition (EEMD), Quantile Regression Analysis (QRA), and Quantile-on-Quantile Regression (QQR) is innovative and effectively addresses the limitations of traditional linear models. These methods capture asymmetric and frequency-dependent relationships, providing a nuanced understanding of fuel demand dynamics.

o The integration of the Marshallian demand framework with nonlinear techniques bridges theoretical and empirical gaps, offering a fresh perspective on consumer behavior.

2. Policy Relevance:

o The findings challenge conventional assumptions about fuel price inelasticity and uniform demand, highlighting the need for targeted subsidies and differentiated pricing policies. This is particularly relevant for Ghana, given its significant fuel price volatility.

o The study’s emphasis on heterogeneity in consumer responses (e.g., petrol as a normal good for middle-income groups but inferior for high-income groups in the long term) provides actionable insights for equitable policy design.

3. Comprehensive Data Analysis:

o The use of monthly data from 2000–2022 ensures robustness, and the decomposition into short-, medium-, and long-term horizons (via IMFs) adds depth to the analysis.

o The validation of QQR results against QRA and the inclusion of 3D visualizations enhance the transparency and credibility of the findings.

Areas for Improvement:

1. Clarification of Methodological Choices:

o EEMD Justification: While EEMD is introduced as a noise-reduction technique, the manuscript could better explain why this method was chosen over alternatives (e.g., wavelet analysis) and how the selected IMFs (1, 5, residual) align with the study’s time horizons.

o Quantile Selection: The rationale for using 7 quantiles (e.g., Q0.05 to Q0.95) should be explicitly tied to statistical power or economic rationale (e.g., income stratification).

2. Discussion of Limitations:

o The exclusion of macroeconomic variables (e.g., exchange rates, global oil prices) and demographic factors (e.g., urbanization, vehicle ownership) is noted, but their potential impact on results could be discussed more critically. For instance, how might exchange rate fluctuations confound the observed price elasticities?

o The non-stationarity of residuals in long-term analyses (e.g., QQR deviations) warrants deeper discussion. Are these deviations statistically significant, and do they affect policy conclusions?

3. Policy Implications:

o The recommendations (e.g., differentiated subsidies, alternative energy investments) are sound but could be more specific to Ghana’s context. For example:

How might the government implement income-tiered pricing in practice?

What barriers exist to adopting electric vehicles/biofuels in Ghana, and how can they be addressed?

o A brief comparison with successful policies in similar economies (e.g., Nigeria, Kenya) could strengthen the policy section.

4. Presentation and Clarity:

o Figures/Tables: Some figures (e.g., 3D plots in Figure 2) are visually dense and could benefit from clearer labels or annotations to highlight key takeaways.

o Terminology: Acronyms (e.g., IMFs, QQR) should be defined at first use, and technical terms (e.g., "inferior good") could be briefly explained for interdisciplinary readers.

General Comments:

• Data Availability: The manuscript states data are "fully available without restriction," but no repository or DOI is provided. Clarifying where data can be accessed would enhance reproducibility.

• References: Some citations lack titles or DOIs (e.g., [6], [22]), reducing traceability. Ensure all references follow journal guidelines.

• Include some of these Key Suggested References:

• https://doi.org/10.1016/j.compeleceng.2021.107295

• https://doi.org/10.3390/en16020642

• https://doi.org/10.3390/su15021222

• https://doi.org/10.3390/su15075739

• https://doi.org/10.1038/s41598-024-57231-7

• https://doi.org/10.1016/j.heliyon.2023.e19387

• https://doi.org/10.1038/s41598-025-85484-3

• https://doi.org/10.1038/s41598-024-61413-8

• https://doi.org/10.3390/su15108264

• https://doi.org/10.1038/s41598-024-83826-1

• https://doi.org/10.3390/su151310736

• https://doi.org/10.3390/en160206425

• https://doi.org/10.1016/B978-0-323-85169-5.00023-X

• https://doi.org/10.1016/j.rineng.2025.105828

• https://doi.org/10.1038/s41598-025-07038-x

• https://doi.org/10.1007/s43621-025-01563-5

The overall evaluations of this manuscript makes a significant contribution to energy economics by combining advanced econometrics with the Marshallian framework to dissect fuel demand elasticities in Ghana. Its findings are robust and policy-relevant, though the impact could be heightened by addressing the above suggestions and accept with major revisions to clarify methodological choices, expand policy discussions, and improve presentation.

**Do you want your identity to be public for this peer review?** For information about this choice, including consent withdrawal, please see our Privacy Policy

Reviewer #1: **Yes: ** Takele Ferede Agajie

---

## [Author Response · Author response to Decision Letter 1]

12 Sep 2025

Every comment has been captured in the "Response to Reviewers" file

---

## [Editor Report · Decision Letter 1]

30 Sep 2025

Dissecting fuel demand elasticities in Ghana: a quantile regression analysis using the Marshallian demand framework

PONE-D-25-34300R1

Dear Dr.

We’re pleased to inform you that your manuscript has been judged scientifically suitable for publication and will be formally accepted for publication once it meets all outstanding technical requirements.

Kind regards,

sunny narayan

Academic Editor

PLOS ONE
---

## [Editor Report · Acceptance letter]

PONE-D-25-34300R1

PLOS ONE

Dear Dr. Takyi,

I'm pleased to inform you that your manuscript has been deemed suitable for publication in PLOS ONE. Congratulations! Your manuscript is now being handed over to our production team.

Kind regards,

on behalf of

Dr. sunny narayan

Academic Editor

PLOS ONE